# FROM MASKS TO WORLDS: A HITCHHIKER'S GUIDE TO WORLD MODELS

## ABSTRACT

This is not a typical survey of world models, it is a guide for those who want to build worlds. We do not aim to catalog every paper that has ever mentioned a "world model". Instead, we follow one clear road: from early masked models that unified representation learning across modalities, to unified architectures that share a single paradigm, then to interactive generative models that close the action-perception loop, and finally to memory-augmented systems that sustain consistent worlds over time. We bypass noisy branches to focus on the core: the generative heart, the interactive loop, and the memory system. We show that this is the most promising path towards world models.

## 1 INTRODUCTION: THE NARROW ROAD TO WORLD MODELS

The term *world model* has been used to describe many different ideas: learned environment simulators for reinforcement learning (Ha & Schmidhuber, 2018; Hafner et al., 2019), agents that integrate learned models with planning (Schrittwieser et al., 2020), and large language models that simulate entire societies (Park et al., 2023). Yet despite hundreds of related works, there is no clear consensus on how to actually build a true world model. In this paper, we take a stance: the path is much narrower than it appears.

A true world model is not a monolithic entity, but a system synthesized from three core subsystems: a generative heart to produce coherent world states, an interactive loop to close the action-perception cycle in real time, and a persistent memory system to sustain coherence over long horizons. The history of the field can be understood as an evolutionary journey to first master these components in isolation, and now, to integrate them. Most works focus on optimizing narrow tasks and drift away from the generative, interactive, and persistent nature required for a true world model.

To make this perspective concrete, we chart the historical evolution of world models as a sequence of five stages, shown in Figure 1. It begins with Stage I: Mask-based Models, which established a universal, token-based pretraining paradigm across modalities. This foundation enabled Stage II: Unified Models, where a single architecture learns to process and generate multiple modalities. The focus then shifts to closing the interactive loop in Stage III: Interactive Generative Models, transforming static generators into real-time simulators. To sustain these simulations over time, Stage IV: Memory and Consistency introduces mechanisms for durable and coherent state representation. Table 1 also summarizes representative models or methods across the four stages.

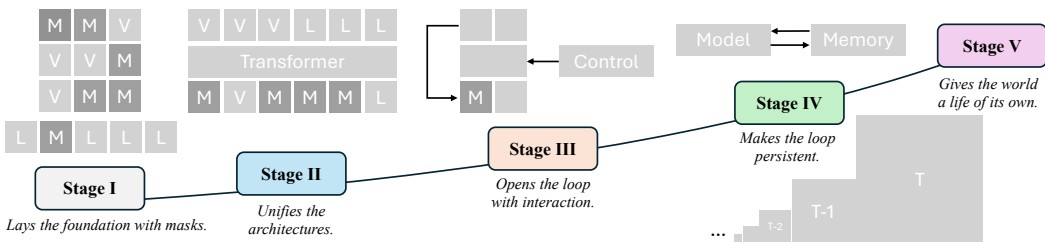

Figure 1: The evolution of world models across five stages.

Table 1: Representative models or methods along the narrow road to world models.

| | |
|---|---|
| **Stage I: Mask-based Models** | |
| **BERT** (Devlin et al., 2019) | Bidirectional masked prediction for representation learning in language. |
| **RoBERTa** (Liu et al., 2019) | Dynamic masking and scale without next-sentence prediction strengthen BERT. |
| **Gemini Diffusion** (DeepMind, 2025) | Reported iterative denoising paradigm at commercial scale for generative language tasks. |
| **BEiT** (Bao et al., 2021) | Image patch masking for representation learning in vision. |
| **MAE** (He et al., 2022a) | High-ratio patch masking with lightweight decoder yields strong visual representations. |
| **MaskGIT** (Chang et al., 2022) | Non-autoregressive parallel masked tokens infilling for efficient image synthesis. |
| **Meissonic** (Bai et al., 2024) | Masked generative transformers achieving high fidelity text-to-image generation. |
| **wav2vec 2.0** (Baevski et al., 2020) | Audio latent features masking for representation learning in speech. |
| **Stage II: Unified Models** | |
| **EMU3** (Wang et al., 2024) | AR-based unified models with a single Transformer for text, image and video. |
| **Chameleon** (Chameleon Team, 2024) | AR-based unified models with a single Transformer for text and image. |
| **VILA-U** (Wu et al., 2024) | Language-prior AR-based unified models for text, image and video. |
| **Janus-Pro** (Chen et al., 2025) | Language-prior AR-based unified models for text and image. |
| **MMaDA** (Yang et al., 2025) | Language-prior mask-based (discrete-style denoising) unified models for text and image. |
| **UniDiffuser** (Bao et al., 2023) | Visual-prior diffusion-based unified models for text and image. |
| **Muddit** (Shi et al., 2025) | Visual-prior mask-based (discrete-style denoising) unified models for text and image. |
| **UniDisc** (Swerdlow et al., 2025) | Mask-based (discrete-style denoising) unified models. |
| **Gemini** (Comanici et al., 2025) | Google's multimodal model in a single system (but not in a single paradigm). |
| **GPT-4o** (Hurst et al., 2024) | OpenAI's multimodal model in a single system (but not in a single paradigm). |
| **Stage III: Interactive Generative Models** | |
| **TextWorld** (Côté et al., 2018) | Parser-based text game environments. |
| **AI Dungeon** (Latitude, 2024) | LLM-driven co-authored narrative with open-ended branching stories. |
| **PVG** (Menapace et al., 2021) | Stepwise playable video game conditioned on user action selection. |
| **PE** (Menapace et al., 2022) | 3D playable environments conditioned on camera and multi-object control. |
| **PGM** (Menapace et al., 2024) | Promptable game model conditioned on semantic-level language control. |
| **GameGAN** (Kim et al., 2020) | GAN-based next frame generation conditioned on actions for 2D games. |
| **Genie-1** (Bruce et al., 2024) | MaskGIT-based next frame generation conditioned on actions for 2D worlds. |
| **Oasis** (Decart et al., 2024) | Open-source Diffusion-based real-time generation conditioned on actions for 3D games. |
| **GameNGen** (Valevski et al., 2024) | Diffusion-based real-time next frame generation conditioned on actions for 3D games. |
| **Genie-2** (Parker-Holder et al., 2024) | Diffusion-based generation conditioned on actions for 3D worlds initialized from images. |
| **Genie-3** (Ball et al., 2025) | Real-time generation conditioned on actions and promptable world events for 3D worlds. |
| **Mineworld** (Guo et al., 2025) | Open-source MaskGIT-based generation conditioned on actions for 3D games. |
| **Matrix-Game-2** (He et al., 2025) | Open-source diffusion-based real-time generation conditioned on actions for 3D games. |
| **World Labs** (World Labs, 2024) | Explorable 3D environments generation from a single image using geometry and depth. |
| **Stage IV: Memory & Consistency** | |
| **RETRO** (Borgeaud et al., 2022) | Improving LMs by conditioning on document chunks retrieved from a large corpus. |
| **MemGPT** (Packer et al., 2023) | OS-inspired virtual memory management framework for LLM workflows. |
| **Transformer-XL** (Dai et al., 2019) | Segment-level recurrence with relative positions for long-context sequence modeling. |
| **Compressive Transformer** (Rae et al., 2019) | Extends Transformer-XL by downsampling old states to retain long-range dependencies. |
| **Mamba** (Gu & Dao, 2023) | Selective state-space model with linear-time recurrence supporting near-infinite context. |
| **FramePack** (Zhang & Agrawala, 2025) | Packs long-frame histories into fixed context with inverted sampling to reduce drift. |
| **MoC** (Cai et al., 2025) | Learnable sparse attention routing that retrieves informative history chunks and anchors. |
| **VMem** (Li et al., 2025a) | Introduces surfel-indexed view memory using 3D surfels to enforce spatial coherence. |

This progression culminates in Stage V: True World Models. This stage is not defined by adding a new component, but by the synthesis of the preceding stages into an autonomous whole. At this threshold, models begin to exhibit the defining properties of persistence, agency, and emergence, moving from engines of prediction to living worlds. By analyzing each stage's key innovations and unsolved challenges, this paper offers a clear and opinionated roadmap from today's components to tomorrow's living worlds.

## 2 WHAT IS A WORLD MODEL?

### 2.1 HISTORICAL AND CONTEMPORARY PERSPECTIVES

The concept of a world model originated in reinforcement learning, where Ha and Schmidhuber (Ha & Schmidhuber, 2018) first proposed learning a latent dynamics simulator for agent planning. This control-oriented view was advanced by systems like Dreamer (Hafner et al., 2019), which learned policies purely through latent imagination, and MuZero (Schrittwieser et al., 2020), which integrated tree-based planning with a learned, abstract model. In parallel, the rise of large-scale generative modeling broadened this definition. With generative agents (Park et al., 2023) and large multimodal systems (Reed et al., 2022), the concept evolved from a predictive simulator for an agent to a rich, generative system that could be an entire interactive world. This has led to the contemporary view of a "world simulator", a term that now informally encompasses three major paradigms: explicit 3D scene generators (World Labs, 2024), passive video generators that go beyond pixels to approximate physical dynamics (Brooks et al., 2024), and interactive games and environments for agents, whether text-based (Niesz & Holland, 1984) or video-based, as exemplified by the Genie series (Bruce et al., 2024; Parker-Holder et al., 2024; Ball et al., 2025).

### 2.2 THE ANATOMY OF A TRUE WORLD MODEL

To bring clarity to these diverse threads, we define a true world model by the three essential subsystems it must integrate, which in turn enable the core properties that define each stage of our evolutionary roadmap. Figure 2 presents the high-level architecture of a true world model, showing how the generative, interactive, and memory subsystems integrate.

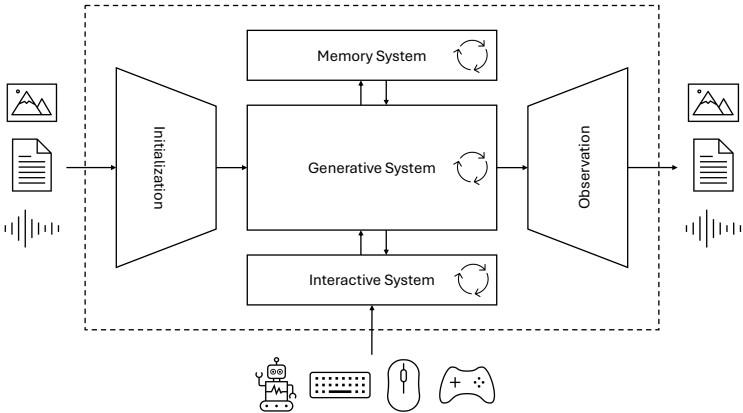

Figure 2: The architecture of a true world model.

**The Generative Heart** ($\mathcal{G}$). The foundation of a world model is its generative heart: a learned model of the world's dynamics and appearance, formally described by the generative process $p_\theta$. It must be able to predict future states, observations, and the task-relevant outcomes.

$$\mathcal{G} = \Big( \underbrace{p_\theta(z_{t+1} \mid z_t, a_t)}_{\text{Dynamics}}, \ \underbrace{p_\theta(o_t \mid z_t)}_{\text{Observation}}, \ \underbrace{p_\theta(r_t \mid z_t, a_t)}_{\text{Reward}}, \ \underbrace{p_\theta(\gamma_t \mid z_t, a_t)}_{\text{Discount/Termination}} \Big)$$

This subsystem, which models state transitions, observations, rewards, and terminations, is the foundation for the property of **Generation**.

**The Interactive Loop** ($\mathcal{F}, \mathcal{C}$). To be more than a passive movie generator, the model must support a closed interactive loop. For partially observable worlds, it requires an *inference filter* ($q_\phi$) for the agent to interpret observations in real-time, and a *policy* ($\pi_\eta$) for it to act upon its understanding of the world, often paired with a value function ($v_\omega$) to evaluate trajectories.

$$\mathcal{F}: \ \underbrace{q_\phi(z_t \mid h_{t-1}, o_t)}_{\text{State Inference}}, \qquad \mathcal{C} = \Big( \underbrace{\pi_\eta(a_t \mid z_t, h_t)}_{\text{Policy}}, \ \underbrace{v_\omega(z_t, h_t)}_{\text{Value}} \Big)$$

This loop is what enables true **Interaction** and **Real-time Adaptation**.

**The Memory System** ($\mathcal{M}$). Finally, to ensure coherence over time, the model needs a memory system that allows past events to inform the future. This is formally captured by a recurrent state, $h_t$, which is updated based on past memory, the current inferred state, and the last action.

$$\mathcal{M}: \quad \underbrace{h_t = f_\psi(h_{t-1}, z_t, a_{t-1})}_{\text{Memory Update}}$$

This component is the basis for the property of **Memory**.

A detailed formalism of each component is provided in Appendix A. This definition clarifies why a system like a Unified Model (Stage II) is a precursor, not a true world model. While it may possess a powerful generative heart, it typically lacks the dedicated interactive loop and explicit memory system required to sustain a persistent, agent-inhabited world.

## 3 STAGE I: MASK-BASED MODELS ACROSS MODALITIES

The first stage in the evolution toward world models is the era of *mask-based modeling*, where a system learns by reconstructing missing or corrupted parts of its input. This paradigm, which can be summarized as mask, infill, and generalize, has proven to be strikingly universal across modalities. It provides a unified way of tokenizing, representing, and pretraining large models, establishing the foundation for all subsequent stages.

### 3.1 LANGUAGE MODALITY

Masked language modeling (MLM) has played a foundational role in modern natural language processing. BERT (Devlin et al., 2019) introduced bidirectional context prediction, where 15% of tokens in each input are randomly replaced with a [MASK] symbol and predicted from surrounding context. SpanBERT (Joshi et al., 2020) refined this approach by masking contiguous spans rather than isolated tokens, improving extraction and reasoning tasks. Sequence-to-sequence variants such as MASS (Song et al., 2019), T5 (Raffel et al., 2020), and BART (Lewis et al., 2019) reformulated MLM as a denoising autoencoding objective. ELECTRA (Clark et al., 2020) improved sample efficiency by replacing the MLM objective with a discriminative replacement-detection task.

Beyond fixed-ratio masking, a line of non-autoregressive work introduces dynamic masking and unmasking through iterative refinement. RoBERTa (Liu et al., 2019) demonstrated that simply optimizing BERT's training recipe with more data and dynamic masking yielded significant gains. Mask-Predict (Ghazvininejad et al., 2019) introduced iterative refinement, re-masking low-confidence tokens over several passes. This concept culminated in discrete diffusion models (Li et al., 2022; He et al., 2022b; Gong et al., 2022; Ou et al., 2024; Sahoo et al., 2024; Shi et al., 2024), which replace fixed masking with a time-indexed noise schedule and train the model to iteratively denoise. As demonstrated by industrial systems like Mercury (Inception Labs et al., 2025) and Gemini Diffusion (DeepMind, 2025), this dynamic denoising paradigm has matured to rival or exceed autoregressive baselines in both quality and inference speed, solidifying the power of masking as a core generative principle (Yu et al., 2025c; Li et al., 2025b).

### 3.2 VISION MODALITY

The masked image modeling (MIM) paradigm extended this principle to perception. Early works established two main branches. For representation learning, BEiT (Bao et al., 2021) and especially MAE (He et al., 2022a) created direct visual analogues to BERT, reconstructing masked tokens or patches to learn powerful features. This spurred a family of related works exploring different reconstruction targets and self-distillation techniques (Xie et al., 2022; Zhou et al., 2021; Wei et al., 2022).

For generative modeling, MaskGIT (Chang et al., 2022) and MUSE (Chang et al., 2023) pioneered the use of masked infilling for high-quality parallel image synthesis. This generative trajectory has recently culminated in models like Meissonic (Bai et al., 2024), which demonstrates that masked

generative transformers can achieve fidelity rivaling large diffusion models while offering superior efficiency and control.

This mask-reconstruct-generalize principle scaled effectively to video. VideoMAE (Tong et al., 2022) and MaskFeat(Wei et al., 2022) showed that high-ratio tube masking was a data-efficient method for learning spatiotemporal representations, confirming that masking could capture not just static scenes but also their dynamics.

### 3.3 OTHER MODALITIES

The universality of the masking paradigm was confirmed by its rapid adoption in other fields. In audio, models like wav2vec 2.0 (Baevski et al., 2020), HuBERT (Hsu et al., 2021), WavLM (Chen et al., 2022), and Audio-MAE (Huang et al., 2022) applied masked prediction to latent speech representations. In 3D domains, Point-BERT (Yu et al., 2022) and Point-MAE (Pang et al., 2023) adapted masking to point clouds. The principle was even extended to structured data with models like GraphMAE (Hou et al., 2022). These successes reinforced masking as a cross-domain general approach to self-supervised learning.

In summary, Stage I established the principle of masking as a universal foundation for representation learning. While this unified the pretraining paradigm, the models themselves remained specialized architectures. The inability of these separate models to form a holistic worldview motivated Stage II: the pursuit of a single, unified architecture.

## 4 STAGE II: UNIFIED MODELS

Stage I established a universal paradigm for representation learning, but the models themselves remained specialists locked within their own modalities. Stage II takes the crucial next step: unifying the models themselves. We define a unified model as a system that processes and generates across different modalities with the shared backbone and the same paradigm. By collapsing modality-specific pipelines, these models simplify scaling, enable powerful cross-modal transfer, and represent the first decisive synthesis on the path toward a true world model.

### 4.1 REPRESENTATIVE WORKS

Leading unified modeling efforts span several trajectories, distinguished by their foundational paradigm. We exclude simple glue models that stitch different paradigms for different modalities, such as using autoregression for text and diffusion for image, as well as models limited to text generation without extending to image generation or other modalities.

**Extending Language Model Pre-training: Language-Prior Modeling**. The dominant trajectory has been to extend the paradigm of autoregressive large language models (LLMs) (Radford et al., 2019; Brown et al., 2020). This began by connecting pre-trained vision encoders to frozen LLMs, as pioneered by BLIP-2 (Li et al., 2023) and popularized by LLaVA (Liu et al., 2023b; 2024), which was built upon LLaMA (Touvron et al., 2023). This approach was pushed further into grounded multimodal reasoning by Kosmos-2 (Peng et al., 2023) and embodied reasoning by PaLM-E (Driess et al., 2023). More recently, systems like the EMU family (Sun et al., 2024; Wang et al., 2024), Chameleon (Chameleon Team, 2024), VILA-U (Wu et al., 2024), and Janus-Pro (Chen et al., 2025) have advanced towards true end-to-end unified generation, creating both text and images within shared token space and unified autoregressive paradigm. In parallel, A notable offshoot of this trend is rooted in mask-based language modeling. LLaDA (Nie et al., 2025) abandons the autoregressive framework and models text through a masked diffusion process with a single Transformer. Its multi-modal extension, MMaDA (Yang et al., 2025), introduces a unified discrete diffusion architecture for text and image, a mixed chain-of-thought fine-tuning strategy, and a policy-gradient RL algorithm (UniGRPO) to unify reasoning and generation across modalities within a single model.

**Extending Vision Model Pre-training: Visual-Prior Modeling**. A parallel effort started from vision-centric foundations, primarily along two paths. The first path built upon latent diffusion models, the foundation laid by Stable Diffusion (Rombach et al., 2022) was later generalized to a unified, joint diffusion process over text and images in models like UniDiffuser (Bao et al., 2023).

The second path built upon the masked image modeling (MIM) paradigm, with models like Muddit (Shi et al., 2025) extending Meissonic (Bai et al., 2024) into a unified discrete diffusion system that produces both images and captions within shared architecture and paradigm. Besides, UniDisc (Swerdlow et al., 2025) trained a unified discrete-diffusion model from scratch for both language and vision modalities.

**Industrial-Scale Unified Systems**. At production scale, Gemini (Comanici et al., 2025) and GPT-4o (Hurst et al., 2024) unify language and vision modalities in a single model, although not in a single paradigm. These demonstrate that unified modeling has transcended research to become a foundational industrial paradigm.

## 4.2 Benefits and Gaps

The primary benefit of Stage II is the reduction of fragmentation, leading to powerful cross-modal transfer and emergent capabilities. This paradigm now underpins productized multimodal interaction at scale, as demonstrated by industrial systems like Gemini (Comanici et al., 2025) and GPT-4o (Hurst et al., 2024). However, despite the impressive progress of language-prior unified models in interactive dialogue, visual-prior unified models for text-to-image and text-to-video remain limited to single-shot synthesis or stepwise editing. They lack the capacity for continuous, real-time closed-loop interaction. Thus while Stage II unified architectures, the creation of truly dynamic and interactive worlds remains an open challenge and motivates Stage III.

## 5 Stage III: Interactive Generative Models

Here, models are no longer static predictors or one-shot generators, but participants in a closed action-perception loop, sustaining interaction through low-latency response and action-conditioned evolution. We define interactive generative models as systems whose outputs are conditioned on streamed inputs or user actions, supported by internal state. We explore this evolution across three distinct domains: language-based, video-based and scene-based.

### 5.1 Language-based Worlds: Interaction as Narrative

Classic interactive fiction (IF) (Niesz & Holland, 1984; Montfort, 2011; Ammanabrolu et al., 2020) established the paradigm of text-driven worlds where players interact through textual descriptions and actions. These took several forms: parser-based games where the player types text commands character by character, choice-based games where the player selects from a set of predefined action options, hypertext-based games where the player clicks on links embedded in the narrative. Choice-based visual novels, such as Memories Off (KID, 2004), exemplify emotionally branching narratives in which player decisions directly affect relationships and endings. These static worlds naturally evolved into benchmarks for artificial intelligence. A significant line of research, supported by platforms like TextWorld (Côté et al., 2018) and Jericho (Hausknecht et al., 2020), was dedicated to training agents that could master them. In these settings, the world was a fixed puzzle to be solved, and the locus of intelligence was the agent who navigated a static world, not the world itself.

A fundamental shift occurred when large language models (LLMs) (Hurst et al., 2024; Comanici et al., 2025) themselves became the world engine. AI Dungeon (Latitude, 2024) pioneered this transition, dynamically generating new narrative branches in response to free-form user prompts. Players could explore unbounded story spaces limited only by imagination and the model's generative capacity. This marked the transition from solving pre-authored worlds to co-creating open-ended ones, envisioning a future where visual novels such as Memories Off (KID, 2004) could be interactively generated, offering unique storylines and relationships for each player.

### 5.2 Video-based and Scene-Based Worlds: Interaction as Experience

Interactive generation in video and spatial domains has progressed from offline frame prediction to real-time, controllable simulation. Early work on world models (Ha & Schmidhuber, 2018) used latent rollouts to "dream" trajectories for policy training, demonstrating the potential of closed-loop simulation. GameGAN (Kim et al., 2020) advanced this idea into a neural game engine, rendering successive frames from user input while implicitly learning game rules from observation. User

control evolved from stepwise action selection in Playable Video Generation (PVG) (Menapace et al., 2021), through 3D scenes with camera and multi-object control in Playable Environments (PE) (Menapace et al., 2022), to natural language prompts in Promptable Game Models (PGM) (Menapace et al., 2024), which enabled semantic-level direction of play.

Building on these conceptual foundations, a decisive trajectory emerged with the Genie series. Genie-1 (Bruce et al., 2024) learned latent action interfaces from Internet-scale videos to create controllable 2D environments. Genie-2 (Parker-Holder et al., 2024) extended this capability to larger, quasi-3D spaces, initialized from a single image and playable via standard controls. Genie-3 (Ball et al., 2025) scaled further, producing real-time text-to-world experiences at 720p and 24 fps with minutes of coherent play, a marked shift from passive video generation to active interaction.

Community and industrial efforts soon followed. Systems such as Oasis (Decart et al., 2024), GameNGen (Valevski et al., 2024), Mineworld (Guo et al., 2025), and Matrix-Game (He et al., 2025) demonstrated *real-time* open environments with emergent physics and streaming diffusion. For a comprehensive overview, see the survey by Yu et al. (2025b).

Beyond frame synthesis, scene-based approaches emerged. World Labs (World Labs, 2024) proposed large world models that generate explorable 3D environments from a single image, enabling interactive navigation through generated geometry and depth rather than sequential video.

Taken together, these advances trace a trajectory from offline video generators to real-time, action-conditioned world simulators. They ultimately transform generative models into engines of interactive human experiences.

## 5.3 CHALLENGES

Despite the leap to real-time interaction, sustaining long-horizon consistency remains unsolved. Two paradigms illustrate the tension: explicit scene generators like NeRFs and Gaussian Splatting (*e.g.*, World Labs) offer stable 3D navigation environments but depend on explicit spatial modeling; implicit frame-by-frame generators offer flexibility but are brittle, prone to losing context and hallucinating objects, especially over extended play. The Genie series highlights this tradeoff: from Genie-1's short 16-frame memory (Bruce et al., 2024), to Genie-2's object permanence (Parker-Holder et al., 2024), to Genie-3's few minutes of coherence (Ball et al., 2025), progress is clear yet far from persistence. At the object level, implicit video models rely on KV caches or control signals to maintain identity, while explicit 3D approaches embed spatial location directly but still struggle with dynamic elements, as explored in 4D Gaussian Splatting. These challenges reveal a deeper gap: the reactive action–perception loop enables interaction, but without dedicated memory and state management, it cannot sustain persistent worlds, which is the central theme of Stage IV.

## 6 STAGE IV: MEMORY AND CONSISTENCY

A world model that acts without memory is reactive yet forgetful. This stage aims to endow models with mechanisms that sustain coherent state across long horizons. The central question emerges: can world models not only generate but also sustain coherent histories, preserve identities, and resist drift? We organize this section around three questions: where to anchor memory, how to extend its span and capacity, and how to govern it to preserve consistency.

## 6.1 EXTERNALIZED MEMORY

Retrieval augments parametric models with non-parametric, often editable, knowledge stores. Early explorations such as Neural Turing Machines (Graves et al., 2014), Differentiable Neural Computers (Graves et al., 2016), and End-to-End Memory Networks (MemN2N) (Sukhbaatar et al., 2015) first explored learnable read–write memory slots. While conceptually groundbreaking, their complexity gave way to more pragmatic, decoupled designs. kNN-LM (Khandelwal et al., 2019), REALM (Guu et al., 2020), and RAG (Lewis et al., 2020) showed that conditioning on retrieved passages could dramatically expand effective context while keeping knowledge traceable and updatable. DPR (Karpukhin et al., 2020) and RETRO (Borgeaud et al., 2022) scaled this approach to dense retrievers and trillion-token databases, rivaling far larger dense LMs while providing traceable and updatable evidence.

Beyond simple retrieval, research has sought to make memory more scalable and dynamic. Product Key Memory (PKM) (Lample et al., 2019) supported massive lookup capacity through factorized keys; MemGPT (Packer et al., 2023) reframed LLMs as operating systems with explicit virtual memory management; LONGMEM (Wang et al., 2023) extends KV caches beyond 65k tokens through decoupled readers; and From RAG to Memory (Gutiérrez et al., 2025) extended retrieval into continual learning, enabling dynamic knowledge updates without retraining. These systems collectively signal a shift from retrieval as a tool to memory as a co-evolving substrate.

## 6.2 EXTENDING CAPACITY AND SPAN

Parallel efforts seek to build persistence directly into the architecture, moving beyond fixed-length attention windows. Within Transformers, Universal Transformer (Dehghani et al., 2018) introduced depth-wise recurrence; Transformer-XL (Dai et al., 2019) propagated segment states across windows; the Compressive Transformer (Rae et al., 2019) down-sampled older activations. Subsequent designs such as the Memorizing Transformer (Wu et al., 2022) and Recurrent Memory Transformer (RMT) (Bulatov et al., 2022) attached associative key–value stores or persistent memory tokens, reaching million-token horizons in practice; Infini-attention (Munkhdalai et al., 2024) added a compressive long-term path for unbounded streaming. In parallel, Perceiver-AR (Hawthorne et al., 2022) introduced a latent cross-attention bottleneck, compressing long inputs into a compact representation and enabling autoregression over 100k tokens across text, images, and music. Together, this line of work represents a reformist trajectory that extends attention through recurrence and compression.

A more radical line argues that persistence requires abandoning quadratic attention entirely. Structured state-space and linear-time models such as S4 (Lu et al., 2023), Mamba (Gu & Dao, 2023), and RetNet (Sun et al., 2023) replace attention with recurrent state updates that achieve linear complexity and thereby, in principle, support infinite context. Precursors such as Linear Transformers (Katharopoulos et al., 2020), together with more recent variants such as Hyena (Poli et al., 2023), pointed in this direction with kernels and long-range convolutions. Together, this line of work represents a revolutionary trajectory that abandons attention in favor of continuous dynamical systems.

Scaling strategies and engineering refinements extend these capacities further. LongNet (Ding et al., 2023) employs dilated attention for billion-token contexts; Ring Attention (Liu et al., 2023a) distributes computation across devices for million-token horizons; LSSVWM (Po et al., 2025) adapts state-space updates for long causal video generation. Practical techniques such as ALiBi (Press et al., 2021), LongLoRA (Chen et al., 2023), and StreamingLLM (Zeng et al., 2024) retrofit long-context ability into existing models. Together, this line of work represents a pragmatic trajectory that extends persistence through scaling strategies and engineering refinements.

Ultimately, these three trajectories, reformist, revolutionary, and pragmatic, converge on the same goal: to achieve genuine continuity, creating models that can read a book, watch a film, or play for hours without losing the thread.

## 6.3 REGULATING MEMORY FOR CONSISTENCY

Persistence without discipline degenerates into drift. The nature of this challenge depends critically on the underlying world representation, which has largely followed two paradigms: implicit 2D video frames and explicit 3D scenes.

In implicit, autoregressive video models, the primary challenge is preventing two entangled failures: forgetting, where early content fades, and drifting, where errors compound. Efforts to mitigate one often aggravate the other (Zhang & Agrawala, 2025). The Genie series highlights this progression: Genie-1 (Bruce et al., 2024) suffers from short memory and drifts after only a few frames; Genie-2 (Parker-Holder et al., 2024) introduces object permanence and sustains coherence for about a minute; Genie-3 (Ball et al., 2025) reaches emergent multi-minute consistency. This underscores a broader challenge: autoregressively generating an environment is fundamentally harder than producing a pre-rendered video, since small inaccuracies accumulate over time. To tackle this, FramePack (Zhang & Agrawala, 2025) uses keyframe anchoring and context compression; Self-Forcing (Huang et al., 2025) and CausVid (Yin et al., 2025) impose stronger causal constraints; Context-as-Memory (Yu et al., 2025a) retrieves overlapping past frames to stabilize long video rollouts, and Mixture

of Contexts (MoC) (Cai et al., 2025) learns sparse routing policies that focus attention on salient history.

Conversely, explicit 3D representations that built upon generative assets from models like Trellis (Xiang et al., 2025), or TripoSG (Li et al., 2025c), inherently provide strong spatial consistency. Here, the challenge shifts to representing dynamic changes and long-term object states. Methods like WorldMem (Xiao et al., 2025), geometry-grounded spatial memory (Wu et al., 2025) and surfel-indexed view memory (VMem) (Li et al., 2025a) leverage this explicit 3D structure to maintain a coherent world state over time, including dynamic representations that capture evolving geometry and supporting revisitations across long horizons. Beyond perceptual consistency, maintaining logical and factual coherence in reasoning remains crucial, addressed by techniques that learn to critique their own outputs (Asai et al., 2024).

The overarching lesson is that longer context alone is insufficient. Consistency emerges from explicit policies over memory: what to write, what to retrieve, how to update, and when to forget.

### 6.4 SUMMARY

Stage IV reframes generation as stateful computation. Externalized memory makes knowledge editable. Architectural persistence makes it durable. Consistency policies make it reliable. At production scale, multimodal systems such as Gemini (Comanici et al., 2025) and Claude (Anthropic, 2024) extend these ideas, sustaining million-token contexts across text, audio, and video and coupling long horizons with reasoning for agentic workflows.

A deeper question remains. Are elaborate memory systems fundamental solutions, or are they sophisticated workarounds for the current constraints of hardware and data? The existence of models with massive, brute-force context windows suggests that some memory problems might simply dissolve with sufficient scale, much like how larger models unlocked emergent abilities. Similarly, consistency failures may also stem from limited data diversity or flaws in the data itself, such as contradictory or erroneous text and videos that are only a few seconds long. The answer will determine whether persistence in world models emerges as a natural property of scale, or as the product of carefully engineered memory discipline. When we ask if world models can dream consistently, the answer we seek is not just an engineering target, but a deeper understanding of the interplay between architecture, scale, and data.

## 7 STAGE V: TOWARDS TRUE WORLD MODELS

The preceding stages constructed the necessary components: a universal generative paradigm (I), a unified architecture (II), a real-time interactive loop (III), and a persistent memory system (IV). Stage V is not the addition of another component, but the synthesis of these parts into a cohesive, autonomous whole. A true world model is not merely a sophisticated simulator controlled by a user; it is a self-sustaining computational ecosystem. Its defining properties are not just programmed but emergent. We show that this synthesis gives rise to three defining properties: Persistence, Agency, and Emergence. More details can be found in Appendix B.

## 8 CONCLUSION: BUILDING LIVING WORLDS

This paper has charted a narrow road: a logical progression from the universal paradigm of masking to the threshold of a new reality. We have argued that this path is defined by the sequential mastery of three fundamental capabilities: unified generation, real-time interaction, and persistent memory. These are not ends in themselves, but the necessary foundations for worlds that can truly be called living worlds that persist with their own history, that are inhabited by goal-directed agents, and that give rise to unforeseen emergence.

The pursuit of isolated benchmarks for static tasks is a detour. The true frontier lies in embracing the architectural and theoretical commitments required to build these self-sustaining computational ecosystems. Therefore, the great choice ahead is whether we build worlds as mere tools for entertainment and escapism, or as scientific instruments for comprehending our complexity. The narrow road we have charted leads to this horizon: a future where we forge not just better models, but new mirrors in which to see ourselves.

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

# APPENDIX

## A    A FORMALIZATION OF THE THREE SUBSYSTEMS

This appendix provides a detailed breakdown of the components formalized in Section 2. We consider a standard partially observable Markov decision process (POMDP) formulation where at each timestep $t$, an agent takes an action $a_t$, receives an observation $o_t$, and a reward $r_t$. The world terminates based on $\gamma_t$. The model maintains a latent belief state $z_t$ and a deterministic memory state $h_t$.

**The Generative Heart ($\mathcal{G}$).**    This subsystem models the world's underlying generative process and comprises three components:

- **Dynamics Model** $p_\theta(z_{t+1} \mid z_t, a_t)$: Predicts the next latent state given the current state and an action. This is the core of the model's ability to "dream" futures.

- **Observation Model** $p_\theta(o_t \mid z_t)$: Maps a latent state back to a sensory observation (*e.g.*, a video frame), grounding the latent space in perceptible reality.

- **Outcome Model** $p_\theta(r_t, \gamma_t \mid z_t, a_t)$: Predicts task-relevant outcomes like rewards and termination signals from the latent state.

**The Interactive Loop ($\mathcal{F}, \mathcal{C}$).**    This subsystem enables a closed-loop exchange between an agent and the world model. It consists of:

- **Inference Model (Filter)** $q_\phi(z_t \mid h_{t-1}, o_t)$: Infers the current latent belief state $z_t$ from the new observation $o_t$ and past memory $h_{t-1}$.

- **Control Model (Policy & Value)** $\pi_\eta(a_t \mid z_t, h_t), v_\omega(z_t, h_t)$: The policy selects the next action based on the current belief and memory, while the value function estimates future outcomes, guiding the policy.

**The Memory System ($\mathcal{M}$).**    This subsystem ensures long-horizon coherence. It has one core component:

- **Memory Update Model** $h_t = f_\psi(h_{t-1}, z_t, a_{t-1})$: Updates the deterministic memory state based on the previous memory, the inferred state, and the last action, creating a persistent representation of history.

This component-based formalization provides a unified lens through which to view the historical evolution of the field, from early control-oriented models that focused on specific components (*e.g.*, Ha & Schmidhuber (2018)) to modern generative systems that aim to integrate them all. It forms the analytical foundation for the five-stage roadmap presented in this paper.

## B    STAGE V: TOWARDS TRUE WORLD MODELS

### B.1    THE THRESHOLD: PERSISTENCE, AGENCY, AND EMERGENCE

A true world model ceases to be a program one runs, but a world one enters. Its defining properties are:

- **Persistence**: The world's state and history exist independently of any single user session, accumulating consequence over time. It has a past that can be revisited and a future that unfolds continuously. This is the ultimate fulfillment of the Memory System ($\mathcal{M}$), transforming the property of Memory into an enduring reality.

- **Agency**: The world is inhabited by multiple, goal-directed agents (human or AI) that interact within a shared context. This property is enabled by the Interactive Loop ($\mathcal{F}, \mathcal{C}$), elevating the properties of Interaction and Adaptation into a multi-agent society.

- **Emergence**: The world's macro-level dynamics arise from the micro-level interactions of its agents and underlying rules, rather than being explicitly scripted. The model becomes a crucible for discovering unforeseen social structures, behaviors, and causal chains. This is the critical synthesis that occurs only when the Generative Heart ($\mathcal{G}$), Interactive Loop ($\mathcal{F}, \mathcal{C}$), and Memory System ($\mathcal{M}$) operate in unison over time.

## B.2 THE FRONTIER: THREE DEFINING CHALLENGES

The path to this threshold is defined by three fundamental, unsolved research problems. These are not merely technical hurdles, but grand challenges that constitute the frontier of the field.

**The Coherence Problem (Evaluation).** For conventional models, fidelity is measured against external ground truth. A true world model, however, writes its own history. The challenge is to evaluate the "truth" of a self-generating reality: to formalize and measure its internal logical, causal, and narrative coherence, and to define what it means for such a world to be consistent.

**The Compression Problem (Scaling).** An ever-growing history risks computational collapse. The challenge is to learn causally sufficient state abstractions that preserve consequence while discarding noise, approaching the information-theoretic bounds of predictive representation. Yet even with abstraction, long-horizon dynamics may be computationally irreducible, forcing us to treat world models not only as engineered systems but as objects of scientific observation.

**The Alignment Problem (Safety).** An autonomous, persistent world model is a technology with profound societal implications. The alignment challenge for a true world model operates on two distinct levels. At its base, the model itself can be viewed as a single environment whose generating process must align with human values. However, the complexity arises when this model becomes the substrate for a multi-agent society. The alignment problem then becomes squared: it requires aligning not only the world's underlying laws (the substrate), but also the emergent, unpredictable dynamics of the agents interacting within it. This is the harder challenge, distinguishing a true world model from a mere single environment simulator.

## B.3 THE HORIZON: FROM SIMULATOR TO SCIENTIFIC INSTRUMENT

The journey detailed in this paper, from masks to worlds, has been about forging a new kind of technology. Yet, the ultimate promise of a true world model lies beyond its function as a simulator for entertainment or training.

Once a world model crosses the threshold of persistence, agency, and emergence, it transforms from a technological artifact into a new kind of scientific instrument. It becomes a computational crucible for running experiments on complex adaptive systems such as economies, cultures, and cognitive ecosystems that are impossible to conduct in reality.

The quest for true world models, therefore, is not merely an engineering endeavor. It is a pursuit of the ultimate tool for understanding complexity itself. The narrow road leads here: to a future where we build worlds not to escape our own, but to comprehend it.

## C  THE USE OF LARGE LANGUAGE MODELS

During the preparation of this paper, large language models were used only for language polishing and minor editing. All research ideas, methods, and experimental results were carried out entirely by the human authors.

