# OpenReview forum: "From Masks to Worlds: A Hitchhiker’s Guide to World Models"
_ICLR.cc/2026/Conference — ICLR 2026 Conference Withdrawn Submission_

### Official Review · Reviewer_oaB5 · 2025-10-14

**Soundness:** 3
**Presentation:** 4
**Contribution:** 2
**Rating:** 4
**Confidence:** 4

**Summary:**

The paper is a position piece that defines true world models as systems integrating three parts: a Generative Heart (dynamics/observations/outcomes), an Interactive Loop (online inference + policy/value for closed-loop control), and a Memory System (deterministic state that preserves long-horizon coherence). It frames progress as a five-stage path: (I) cross-modal mask-based pretraining; (II) unified, single-paradigm multimodal models (arguing against mixing paradigms across modalities); (III) interactive generative models that respond to user actions in real time; (IV) memory & consistency mechanisms that go beyond longer windows to explicit write/retrieve/update/forget policies; and (V) synthesis, where systems exhibit persistence, agency, and emergence. A POMDP formalization distinguishes a stochastic belief state ztz_tzt​ from a deterministic memory hth_tht​, clarifying interfaces among components. The paper highlights the practical tension between implicit video generators (flexible but drift-prone) and explicit 3D scene representations (spatially stable but harder for dynamics). Its contribution is conceptual: an opinionated roadmap and a research agenda focused on coherence metrics, information-efficient compression, and alignment at both the substrate and multi-agent levels—rather than a new algorithm or empirical benchmark.

**Strengths:**

S1. Clear systems decomposition. The G/F–C/M split is clean, grounded in  the engineering sense, and maps well onto existing families (world-model RL, masked/diffusion LMs, long-context architectures). It gives practitioners hooks for interfaces between perception, control, and persistence rather than treating “world model” as an amorphous buzzword.

S2. Useful, opinionated curation. The five-stage roadmap filters noise and highlights a plausible integration path: masks → unified single-paradigm modeling → interactive generation → engineered memory/consistency. The insistence on single-paradigm unification is a nontrivial stance that many teams will find clarifying, even if they disagree.

S3. Memory as policy (for write / retrieve / update / forget), not just scale. The argument that longer windows (Ring/Infini/RetNet/etc.) are necessary but insufficient, and that explicit consistency policies (e.g., keyframe anchoring as in FramePack, causal constraints like Self-Forcing, retrieval-stabilized rollouts, sparse routing via MoC) are required, feels accurate and actionable for long-horizon video/scene generation.

**Weaknesses:**

W1. The $z_t$ /$h_t$ split may not reflect what large models actually do

Modern sequence models often blur belief and memory inside a huge hidden state. For sufficiently large SSM/retentive architectures, a clean factorization into stochastic $z_t$ and deterministic $h_t$ may be neither necessary nor learnable.

If end-to-end models with a single recurrent state achieve persistence + agency metrics, the explicit two-state formalism looks like an implementation detail, not a principle.

W2. Limited conceptual novelty. The manuscript largely reorganizes existing lines of work (masked pretraining → unified multimodality → interactive generation → memory/retrieval) into a five-stage narrative. Comparable world-model surveys/position papers already dissect the field along similar axes (generative dynamics, partial observability/inference, control, memory/persistence). The proposed G/(F,C)/M split mirrors standard state-space/POMDP formulations and long-context design tropes, offering little that is theoretically new. The paper does not introduce a distinctive taxonomy, formal criterion for “true world model,” or falsifiable predictions that would differentiate this roadmap from prior framings. The Stage-V properties (persistence/agency/emergence) are aspirational descriptors, not operational definitions with measurable thresholds.

**Questions:**

Under a fixed real-time latency and compute/memory budget (FLOPs, activation RAM, bandwidth), how should resources be optimally allocated across the **filter** $q_\phi$, **policy/value** $\pi_\eta, v_\omega$, the **generative heart** (dynamics/observation), and **memory** (write/retrieve/update/forget) to maximize long-horizon coherence (e.g., identity permanence, loop-closure accuracy), and what empirical **marginal return curves** justify that allocation versus simply enlarging the context window?

---

### Official Review · Reviewer_dE2L · 2025-10-27

**Soundness:** 2
**Presentation:** 2
**Contribution:** 2
**Rating:** 4
**Confidence:** 4

**Summary:**

This paper provides a perspective on the development of “world models” in AI, defining a true world model as a system with three core components: a generative heart, an interactive loop, and persistent memory. The authors propose a five-stage evolutionary taxonomy, from mask-based models and unified architectures to interactive simulations, memory mechanisms, and the final synthesis into a cohesive system exhibiting persistence, agency, and emergent behavior. They argue that most existing work focuses on narrow tasks, neglecting the integrated, generative, and interactive nature required for a true world model.

**Strengths:**

1. The paper presents a conceptual framework for understanding world models, breaking them down into several core components. This decomposition can serve as a useful guide for designing next-generation world models.

2. It covers a broad range of approaches and bridges multiple research subfields, offering a unifying perspective that can inform and inspire future work.

3. The paper raises important conceptual questions about what defines a “true world model,” encouraging the community to consider key properties such as persistence, agency, and emergence.

**Weaknesses:**

1. A key limitation of the paper lies in the lack of conceptual clarity and justification for its central construct — the notion of a “world model.” While the authors propose that a true world model consists of three subsystems (generative, interactive, and persistent) and trace its evolution across several stages, it remains unclear why these specific components or phases are necessary or sufficient. The definition seems partly descriptive and post-hoc, built around existing trends in generative and interactive modeling, rather than derived from a rigorous theoretical or empirical grounding. Therefore, I find it a bit difficult to understand how this taxonomy distinguishes itself from prior frameworks.

2. The framing builds upon existing well-known trends (foundation models → multimodal models → interactive agents → memory systems). While the synthesis is insightful, it might be viewed as a retrospective taxonomy rather than a forward-looking technical innovation.

3. Several models (e.g., Gemini, GPT-4o, or Mamba) exhibit features spanning multiple stages—combining unified modeling, interactive generation, and memory mechanisms—making strict classification ambiguous.

4. Another key limitation of the paper is the lack of empirical evaluation or benchmarks to support the proposed division of world models into distinct stages. Without experiments, metrics, or case studies demonstrating the usefulness or validity of this taxonomy, it remains largely descriptive and theoretical, making it difficult to assess its practical relevance or impact on model design.

**Questions:**

1. In Line 129, the authors present an architecture for what they describe as “a true world model.” However, this concept appears subjective and lacks quantitative grounding. It would be helpful if the authors could clarify the criteria or evaluation metrics that determine when a system qualifies as a “true world model,” as well as provide more insight into the motivation and rationale behind designing such an architecture. In short, how do the authors define a “true world model,” and why are the three subsystems (generative, interactive, persistent) considered necessary and sufficient?

2. The four-stage taxonomy implies a linear or chronological evolution of model design, whereas many of these paradigms overlap in time and share underlying architectures. For example, mask-based (e.g. BERT) and memory models (e.g. Transformer-XL) continue to co-evolve rather than forming distinct sequential stages. It will be great if the authors could clarify how they account for the concurrent development and cross-pollination of these approaches?

3. How would one empirically determine if a system has reached a particular “stage” in the proposed evolutionary framework?

4. Stage V is described as a synthesis of previous components into a “self-sustaining computational ecosystem,” with properties such as Persistence, Agency, and Emergence. However, it is unclear how they are measured, instantiated, or distinguished from earlier stages. What concrete criteria or mechanisms define Stage V, and how does it differ operationally from Stage IV?

---

### Official Review · Reviewer_RwiZ · 2025-10-30

**Soundness:** 1
**Presentation:** 1
**Contribution:** 1
**Rating:** 0
**Confidence:** 3

**Summary:**

This paper tries to establish an evolutionary development of different methods that can be categorized under the umbrella term world models, with a focus on transformer-based generative models and reinforcement learning world models. The authors propose five evolutionary stages: (I) generative models trained with masked tokens, (II) multimodal generative models trained with masked tokens, (III) interactive models, (IV) models with memory, and (V) "true world models." For the first four stages, the authors list representative papers and briefly summarize their contributions.

**Strengths:**

The paper addresses an ambitious and timely question: how the notion of "world models" has evolved across subfields of machine learning. The attempt to synthesize diverse research under a unified conceptual framework is valuable, given the growing overlap between generative modeling, reinforcement learning, and multimodal systems.

**Weaknesses:**

- Much of the text alternates between short, surface-level paper summaries and highly speculative or poetic claims. For instance, the authors describe models that "move from engines of prediction to living worlds" (lines 104-106), call a "true world model" a "self-sustaining computational ecosystem" (lines 468-469), and end with a reflection on "forging new mirrors in which to see ourselves" (lines 483-486). These statements are imaginative but feel disconnected from any clear technical grounding. They read more like visionary speculation than scientific argument.
- The paper lacks a coherent rationale for the proposed evolutionary framework. The categorization appears historically inconsistent, for example, Stage II includes work from 2019, while Stage III cites papers from the 1980s and early 2000s, making it unclear in what sense these stages represent a chronological or conceptual evolution.
- Several categorizations also seem arbitrary. The definition of what constitutes a "world model" is not applied consistently, and given the authors' stated requirements, one could argue that many existing models already satisfy these criteria.
- The paper's most novel component, the description of the "true world model" stage, is relegated to the appendix, leaving the main text without a clear contribution or takeaway.

**Questions:**

- Could the authors clarify why language models such as BERT are categorized as Stage I, while ChatGPT-4o is placed in Stage III?
- Why are Dreamer world models classified under Stage III rather than Stage IV, given that they include an LSTM-based memory component that meets the paper's own definition of memory?
- If the discussion of "true world models" represents the conceptual culmination of the proposed evolution, why is it presented only in the appendix?

---

### Official Review · Reviewer_TWe8 · 2025-10-31

**Soundness:** 3
**Presentation:** 2
**Contribution:** 3
**Rating:** 6
**Confidence:** 3

**Summary:**

This is a (very opinionated) roadmap paper rather than a survey or system paper. It argues that "true world models" arise from synthesizing three subsystems: a Generative Heart (dynamics, observation, outcome models), an Interactive Loop (filter/inference, policy/value), and a Memory System (persistent state). The paper organizes the field into five stages: (1) mask-based pretraining across modalities, (2) unified multimodal models in a single paradigm, (3) interactive generative models that close the action-perception loop, (4) memory and consistency mechanisms for long horizons, and (5) "true world models" exhibiting persistence, agency, and emergence.

**Strengths:**

- The main contribution is a structured framework with POMDP-style formalism and three frontier problems (coherence, compression, alignment) for tracking progress. It's purely conceptual with no new benchmark or model, but it's helpful for organizing current work and pinning down what we actually mean by "world model."
- The unifying vocabulary and diagrams help connect disparate fields like RL, video generation, interactive games, and 3D scene generators that often talk past each other. The authors are pretty opinionated against "bag-of-benchmarks" and "glued paradigms" approaches. They insist that real-world models need closed-loop generation and a persistent state. Even if I don't personally fully agree with all the beliefs, I think this paper is a pleasure to read.

**Weaknesses:**

- Stage boundaries are somewhat normative. Several recent systems occupy hybrids (e.g., unified front-ends with heterogeneous generators), and the paper's "single-paradigm" preference could be premature without controlled ablations showing it's actually better.
- Evaluation remains abstract. The Coherence Problem is posed compellingly, but there's no concrete metric suite offered (like causal-graph consistency, entity-lifecycle preservation, or narrative-logic audits). The Safety/Alignment section is particularly high-level, with multi-agent alignment specifics (substrate-policy vs emergent-policy interventions) not operationalized at all.

**Questions:**

### Q1

The POMDP factorization is introduced but never fully written as a joint distribution $p_θ(o_{1:T}, r_{1:T}, z_{1:T} | a_{1:T})$. Without this, it's unclear how the Generative Heart, Interactive Loop, and Memory System compose mathematically.

**Action:** Explicitly derive and display the joint factorization, showing how each subsystem $(G, F/C, M)$ contributes to the full generative process. This would anchor the framework in probabilistic modeling rather than conceptual analogy.

### Q2

The paper defines $γₜ$ as termination, but reuses it inside $p_θ(r_t, γ_t | z_t, a_t)$. Typically, $γₜ ∈ {0,1}$ is a discount or termination mask, but this usage could confuse readers into thinking it's a latent.

**Action:** Clarify whether γₜ is sampled or deterministic?

### Q3

Stage V (True World Models) introduces these as qualitative thresholds, but no quantitative or operational test distinguishes "persistent" from "memory-augmented" systems.

**Action:** Propose operational metrics or diagnostics, e.g., persistence length (steps of state continuity), causal cycle closure, or mutual information between successive memory states. This helps to make these concepts measurable and falsifiable.

---

### Note · Authors · 2025-11-12

I have read and agree with the venue's withdrawal policy on behalf of myself and my co-authors.